# Analysis of Pharmaceutical Companies’ Social Media Activity during the COVID-19 Pandemic and Its Impact on the Public

**DOI:** 10.3390/bs14020128

**Published:** 2024-02-09

**Authors:** Sotirios Gyftopoulos, George Drosatos, Giuseppe Fico, Leandro Pecchia, Eleni Kaldoudi

**Affiliations:** 1European Alliance for Medical and Biological Engineering and Science, 3001 Leuven, Belgium; gfico@lst.tfo.upm.es (G.F.); l.pecchia@warwick.ac.uk (L.P.); kaldoudi@med.duth.gr (E.K.); 2Institute for Language and Speech Processing, Athena Research Center, 67100 Xanthi, Greece; 3Life Supporting Technologies, Universidad Politécnica de Madrid, 28040 Madrid, Spain; 4School of Engineering, University of Warwick, Coventry CV4 7AL, UK; 5Department of Engineering, Università Campus Bio-Medico di Roma, 00128 Rome, Italy; 6School of Medicine, Democritus University of Thrace, 68100 Alexandroupoli, Greece

**Keywords:** COVID-19 pandemic, pharmaceutical companies, emotion analysis, public impact, social media

## Abstract

The COVID-19 pandemic, a period of great turmoil, was coupled with the emergence of an “infodemic”, a state when the public was bombarded with vast amounts of unverified information from dubious sources that led to a chaotic information landscape. The excessive flow of messages to citizens, combined with the justified fear and uncertainty imposed by the unknown virus, cast a shadow on the credibility of even well-intentioned sources and affected the emotional state of the public. Several studies highlighted the mental toll this environment took on citizens by analyzing their discourse on online social networks (OSNs). In this study, we focus on the activity of prominent pharmaceutical companies on Twitter, currently known as X, as well as the public’s response during the COVID-19 pandemic. Communication between companies and users is examined and compared in two discrete channels, the COVID-19 and the non-COVID-19 channel, based on the content of the posts circulated in them in the period between March 2020 and September 2022, while the emotional profile of the content is outlined through a state-of-the-art emotion analysis model. Our findings indicate significantly increased activity in the COVID-19 channel compared to the non-COVID-19 channel while the predominant emotion in both channels is joy. However, the COVID-19 channel exhibited an upward trend in the circulation of fear by the public. The quotes and replies produced by the users, with a stark presence of negative charge and diffusion indicators, reveal the public’s preference for promoting tweets conveying an emotional charge, such as fear, surprise, and joy. The findings of this research study can inform the development of communication strategies based on emotion-aware messages in future crises.

## 1. Introduction

On 11 March 2020, Dr. Tedros Adhanom Ghebreyesus, the Director-General of WHO, characterized COVID-19 as a pandemic, signifying the beginning of an era of global turmoil with many casualties that classified it amongst the deadliest pandemics in modern history [1]. Throughout this period, the public was bombarded with unverified information, leading to an “infodemic” in parallel to the medical pandemic. The scientific community detected in time the unnecessary risks posed by the circulation of unreliable information and fake news [2], while the WHO provided guidelines to flatten the infodemic curve [3] and issued a joint statement with eight other UN organizations urging member states to act against the spread of mis- and disinformation [4].

Online social networks (OSNs) served as the main channel of interaction for the public in times when physical social distancing and lockdowns were necessary to contain the spread of the virus. Although OSNs provided a virus-safe environment for online users to express their emotions and share their opinions and fears, the excessive use of social media took its toll on their emotional state and mental health [5,6,7]. The increased emotional charge and activity on social media, combined with the phenomena of mis- and disinformation, boosted the circulation of myths and diminished the credibility of all actors, even well-intentioned ones [8,9].

Despite the chaotic information landscape, social media were highly deployed by key actors to readily diffuse crucial information. Recent research confirmed the WHO’s commitment to use Twitter, currently known as X, to disseminate messages containing accurate and scientific information [10], while world leaders tweeted informative and encouraging posts in order to boost the morale of citizens [11,12]. Government and health officials also used Twitter as a tool for crisis management to reassure society [13]. Studies prior to the pandemic had already confirmed the key role of social media, especially Twitter, in the diffusion of medical content to professionals and the general public and had highlighted the pros and cons of this new setting [14,15].

The heated activity on OSNs during the pandemic triggered the scientific community’s interest in analyzing posts on social media and examining users’ attitudes towards official guidelines and restrictions. Research works prior to the COVID-19 pandemic confirmed that OSNs can serve as a fruitful ground for health research since state-of-the-art algorithms can extract valuable information regarding public health topics (e.g., surveil-lance, disease tracking, forecasting, etc.) [16,17]. New technologies (e.g., machine learning, AI, IoT, etc.) have proven efficient in analyzing the circulation of messages on social media and have shed light on the different aspects of public health themes [18,19].

A research field that has received great attention from scientists is the sentiment and emotion analysis of posts made during the pandemic as well as the identification of discussion topics on social media. Sentiment analysis was employed to detect the fluctuations of the public’s attitude from positive to negative and vice versa using word-based techniques [20,21,22,23,24]. Other studies conducted a more fine-grained analysis combining sentiment with emotion analysis in order to investigate the dominant emotions and their correlation to external events and various periods [25,26,27,28]. Topic modeling, mainly using the latent Dirichlet allocation algorithm [29], was performed to identify the most debated themes of discussion [23,24,30,31]. Most of these studies retrieved their data from Twitter, implicitly acknowledging the medium’s suitability for providing datasets that can lead to credible societal analyses.

Pharmaceutical companies played a crucial role in the COVID-19 pandemic as they were tasked with the responsibility to develop the highly anticipated vaccines, the first line of defense against the spread of the disease. The infodemic, coupled with the medical pandemic, cast shadows over the integrity of the procedures followed during the development of vaccines. As a result, nine CEOs of the most prominent pharmaceutical companies issued a common statement signing “a historic pledge to continue to make the safety and well-being of vaccinated individuals the top priority in development of the first COVID-19 vaccines” [32]. Despite the phenomena of misinformation that attacked the credibility of science, scientists maintained high levels of trust by society throughout the pandemic [33,34] and citizens expressed their confidence in the process of vaccine development [35].

The aim of this study is to analyze the activity of prominent pharmaceutical companies on Twitter during the pandemic and their interaction with the public to highlight the underlying patterns of communication and identify the characteristics of their discourse. The importance of pharmaceutical companies’ scientific role in the battle against the pandemic, combined with elevated trust in science by society, support our claim that the analysis of these companies’ activity and of their interactions with the public can produce valuable results. This study focuses on COVID-19-related tweets made by pharmaceutical companies, which comprise the COVID-19 communication channel, and compares their characteristics to those of non-COVID-19 tweets, which form the non-COVID-19 communication channel. The objective is to investigate the emotional profile of posts made in English both in the COVID-19 and the non-COVID-19 channels and analyze users’ reactions (i.e., retweets, quotes, and replies) to companies’ messages to assess the impact of the companies’ activity on the emotional state of the public. To the best of our knowledge, this is the first study that focuses on the interaction of prominent pharmaceutical companies with the public during the COVID-19 pandemic, covering a wide period of 31 months that encompasses all waves of the virus.

The research questions (RQs) of this study are as follows:RQ1.What are the differences in the activity and emotional profiles of pharmaceutical companies’ posts in the COVID-19 and the non-COVID-19 communication channels?RQ2.How did users respond to posts in the two channels in terms of diffusion and emotion enhancement through their retweet activity?RQ3.Which emotions were triggered in users by posts in the COVID-19 channel?RQ4.How did users respond to posts with different emotional charges in the COVID-19 channel?

Our analysis can be perceived as a study of knowledge transfer between scientific actors and society examined through the prism of the quadruple helix innovation model [36] and its extension, the quintuple helix innovation model [37]. The quadruple helix innovation model is based on interactions between four actors (i.e., academia, industry, the state, and the media and culture-based public) and describes the ecosystem of sustainable innovation through the perpetual transfer of knowledge between them, while its extension introduces the natural environment as the ecosystem’s fifth actor. This work was conducted within the “PandeVITA” EU project [38], which deployed these innovation models to propose a novel platform for knowledge transfer in times of pandemic.

## 2. Materials and Methods

### 2.1. Dataset Compilation

We chose Twitter as the source of data for our approach since recent scientific reviews confirmed this medium’s fitness for emotion analysis [39,40]. These reviews examined a considerable number of published articles (>40) that studied users’ behavior during the COVID-19 pandemic through different perspectives, mainly utilizing the platform’s data. Additionally, Twitter granted us a research account acknowledging the scientific questions of the study and provided us with unlimited free access to its data through its API at the time when the analysis was conducted (September 2022).

The communication channel between pharmaceutical companies and the public on Twitter involves the accounts of the companies and the users as well as the interactions between them as expressed through the network’s provided actions (i.e., retweet, quote, reply). A list of pharmaceutical companies with worldwide activity was assembled based on a publicly available list on Wikipedia that had been active and updated regularly for more than two decades (accessed on 3 August 2022) [41]. The Twitter accounts of the companies were searched and collected in a process executed by a human to ensure the credibility of the retrieved data: The official web site of each pharmaceutical company was visited through the Google search engine and the advertised Twitter accounts, when available, were collected. The authors also executed a separate query using the company’s name as input to Twitter’s search box and verified accounts that contained the name were included. Finally, the friends’ list of each verified company was examined and the accounts that contained the company’s name and corresponded to other branches (e.g., R&D, marketing, country-specific) were collected. An analogous automated process required the definition of strict criteria for the detection and validation of accounts, a challenging task that could jeopardize the credibility of the data since some steps of the process required flexibility (e.g., the detection of the Twitter account on a web page). 

The activity of these Twitter accounts (i.e., original tweets, retweets, quotes, replies) was retrieved for the period between March 2020 and September 2022 via the Twitter API. The retrieved activity was separated into two channels of communication (i.e., the COVID-19 and the non-COVID-19 channel) using an annotating mechanism provided by Twitter, which labeled each tweet based on its relevance to COVID-19.

### 2.2. Tools and Techniques

The average values of likes, quotes, replies, and retweets were calculated as indicators of the captured information flow [42,43]. The calculations were conducted with the Python programming language and graphs were constructed using modules of Microsoft Excel for Microsoft 365 MSO, Version 2401. Lists of most frequently used words and hashtags were compiled to provide evidence regarding activity content. Table A1 and Table A2 of Appendix A present the lists of words and hashtags both for the tweets circulated by the pharmaceutical companies and the retweets of the users.

Emotion analysis of posts made in English was conducted via the Python programming language implementing a deep learning algorithmic model based on machine learning approaches, specifically recurrent neural networks (RNNs) [44]. The efficiency of the applied model has been established by recent studies in the domains of politics [45], health [46], and the COVID-19 pandemic [30,47,48]. In our approach, we utilized the Ekman classification scheme available in the model, which uses six basic emotions (i.e., anger, disgust, fear, joy, sadness, and surprise) to define the output categories of the classifier. 

The model assigned each tweet a factor of value from 0 to 1 for each emotion. The sum of the respective factors of each emotion denoted the emotional charge of the tweet and, when the sum was less than 1, the remainder was considered neutral. In case the sum of the six factors exceeded the unit, the neutral factor was considered zero and the rest were normalized to a sum of 1. The seven factors (i.e., six for the six emotions and one for the lack of emotions—neutral factor) can be perceived as percentages—probabilities of different emotional charges in each tweet. When a factor was greater than 0.5, the tweet was labeled with the corresponding emotion and was classified accordingly.

## 3. Results

### 3.1. Dataset

The authors assembled a list of 187 companies that operate worldwide and identified 630 related Twitter accounts. Many pharmaceutical companies were associated with multiple accounts as the different departments (e.g., R&D, marketing) or branches of the company (e.g., Europe, USA, India) controlled dedicated feeds. The activity of the pharmaceutical companies in the non-COVID-19 and COVID-19 communication channels is presented in Table 1.

The public’s response to the companies included reactions to the companies’ posts in the form of retweets, quotes, and replies. The public’s reaction is presented in Table 2.

For the emotion analysis, users’ quotes and replies made in English in the COVID-19 channel were collected and are presented in Table 3.

### 3.2. Activity and Emotional Profile of Pharmaceutical Companies’ Posts in the COVID-19 and Non-COVID-19 Communication Channels (RQ1)

Table 1 shows that the ratios of COVID-19-related to non-COVID-19-related original tweets are high for both total posts and posts made in English (i.e., 0.1731 ≈ 1:6 and 0.1777 ≈ 1:6, respectively), highlighting the increased activity of pharmaceutical companies regarding the COVID-19 pandemic. The predominant part of their presence on the social media network involves original posts (69.69% in the non-COVID-19 activity channel and 64.31% in the COVID-19 activity channel), while retweets, replies, and quotes have lower shares, with quotes exhibiting the lowest share in all cases.

Figure 1 and Figure 2 present pharmaceutical companies’ total activity in both the non-COVID-19 and COVID-19 communication channels, respectively, per type of action (original posts, quotes, replies) in the study period. It is evident that the companies exhibited intense activity in the COVID-19 channel during the first months of the pandemic, with a declining tendency in the following months, while traffic in the non-COVID-19 channel followed the reverse course. Based on the most frequently used words and hashtags in Table A1 and Table A2 of Appendix A, the outbursts of activity in the COVID-19 channel during March and May 2021, as seen in Figure 2, can be attributed to a fraud attempt that involved a pharmaceutical company included in the dataset and the rollout of a vaccine. Similarly, the peak in February 2022 is related to the marketing strategy of a certain pharmaceutical company regarding World Cancer Day and its relevance to the COVID-19 pandemic.

The emotional profiles of the pharmaceutical companies in both the non-COVID-19 and the COVID-19 communication channels are presented in Table 4. 

A comparison of the emotions in the two channels does not reveal striking variations in the average values, although there are signs indicative of key emotions. To be specific, the overall joy and surprise disseminated by pharmaceutical companies in the COVID-19 channel is noticeably lower than the corresponding values in the non-COVID-19 channel, while the emotion of fear is increased in the COVID-19 channel. Neutrally charged posts share a lower portion in the non-COVID-19 channel and the emotions of anger, disgust, and sadness have either excessively low values (i.e., <0.01) or exhibit insignificant variations, imposing, thus, risks in terms of extracting credible conclusions.

### 3.3. Users’ Response to Pharmaceutical Companies’ Posts in the Non-COVID-19 and the COVID-19 Communication Channels (RQ2)

Table 2 shows that the public’s response in the COVID-19 channel is significantly greater than in the non-COVID-19 channel. More specifically, the response ratio of the public’s total actions (i.e., original tweets, replies, and quotes) to the pharmaceutical companies’ tweets in the COVID-19 channel is 24.10, while the respective ratio in the non-COVID-19 channel is 4.00, as presented in Table 5, which is strong evidence of the public’s interest in the COVID-19 channel. 

Similar conclusions can be drawn from the average likes, quotes, replies, and retweets made by the public to posts disseminated by the pharmaceutical companies, as presented in Table 6. The rates in the COVID-19 channel are consistently higher than the corresponding rates in the non-COVID-19 channel for every type of post. Pharmaceutical companies’ original posts have an intense triggering effect in both channels, with the highest values being in the COVID-19 channel, where each original tweet produced, on average, 110.35 reactions. Similarly, the average retweet rate of original posts in the COVID-19 channel is the highest (19.95), indicating users’ tendency to circulate any disseminated information. The averages of likes and retweets are greater than the corresponding rates of quotes and replies in both channels, signifying the users’ tendency to express themselves more frequently with effortless reactions.

The response of the public per month is shown in Figure 3 and Figure 4. Figure 3 shows that the public’s non-COVID-19-related activity ranges from 24,817 reactions (August 2021) to 71,714 reactions (November 2021). On the other hand, Figure 4 shows that the public’s COVID-19-related activity ranges from 7618 reactions (September 2022) to 136,668 reactions (April 2021), with an outburst of activity between November 2020 and January 2022. During this period, the users’ response in the COVID-19 channel was more intense than in the non-COVID-19 channel, although the pharmaceutical companies were more active in the non-COVID-19 channel.

Another noteworthy finding is depicted by comparing the results in Figure 2 and Figure 4: although companies were more active in the first months of the pandemic, users exhibited the peak of their activity in the period between March and May 2021. This peak can be attributed to the vaccine rollout strategies in different countries, the announcement of new vaccines (e.g., Sputnik), and studies on candidate substances (e.g., remdesivir), according to the word and hashtag frequencies. Similarly, the enhanced activity in November 2021 can be attributed to the circulation of COVAXIN, the COVID-19 vaccine developed in India. A decrease in users’ reactions is detected from February 2022 onwards.

The response of the public, delivered through retweets, altered the emotional profiles of the channels by promoting tweets with different emotional charges. Table 7 presents the emotional profiles disseminated by posts made by pharmaceutical companies and their enhancement achieved through user retweets, with changes greater than 10^−2^ highlighted with indicative arrows. In the non-COVID-19 channel, our findings indicate that the public tended to promote posts that conveyed fear more than those that conveyed joy. In the COVID-19 channel, the public propagated slightly more emotions of fear and surprise, and marginally more joy, while it refrained from sharing neutrally charged posts. The reactions to anger, disgust, and sadness were found to be low.

Figure 5 shows the average values per month for the emotions of fear, joy, and neutral emotional charge. Figure 5a indicates that joy is one of the two predominant emotions, with values ranging from 0.34 to 0.48 for pharmaceutical companies and from 0.29 to 0.72 for the posts propagated by users. In March 2020, the circulation of promoted posts exhibited its highest peak, which can be attributed to the development of rapid tests for the detection of the virus, as indicated by the most frequently used words, while other peaks in the following months can be attributed to the development of vaccines (e.g., Phase I and II human trials of COVAXIN in June, August, and September 2020), the rollout of vaccines (e.g., SputnikV in February 2021), and trivial diagnostic guidelines (e.g., LungTest in May 2021). However, the graph indicates a relative hesitance of the public to promote pharmaceutical companies’ tweets that express joy, as the values for joy in promoted posts are frequently lower than the corresponding values of the companies’ activity. This remark is also supported by the linear estimation of the pharmaceutical companies’ activity, which exhibits a declining trend (slope m = −2 × 10^−5^), and the corresponding estimator for the users’ promotion, which declines at a greater rate (m = −1 × 10^−4^).

Fear exhibited lower values than joy, as depicted in Figure 5b, ranging between 0.04 and 0.09 for the pharmaceutical companies and between 0.04 and 0.22 for the public’s promotion. The public’s response almost consistently surpassed the fear expressed by the pharmaceutical companies since, for several months, the fear promoted by users showed to be double or even triple compared to the values produced by the companies (e.g., November 2020, July 2021, November 2021, June 2022). Based on the results of the words most frequently used in tweets, these spikes in the public’s fear can be attributed to uncertainty and doubt about the vaccines’ efficacy (e.g., November 2020), regional outbursts of the pandemic (e.g., in India in July 2021), as well as the rollout strategy and the development of new vaccines (e.g., June 2022). The enhancement of fear by users is also confirmed by the slightly upward trend of the linear estimator for the users’ promotion (m = 2 × 10^−5^) while the estimator for the companies’ activity declines (m = −1 × 10^−5^). Furthermore, fear exhibited an inverse behavior compared to the corresponding values for joy (e.g., November 2020, November 2021, May–July 2022), a phenomenon that can be justified by the opposed nature of the two emotions. 

The average values of neutral posts presented in Figure 5c are at the same level as the values of joy, ranging between 0.39 and 0.47 for the companies and between 0.11 and 0.53 for user retweets, and in many cases behave complementarily to the sum of fear and joy. In months when the values of either fear (e.g., July 2021, November 2021) or joy (e.g., August 2020, May 2021) peaked, the corresponding values of neutral posts were lower. The public tended to promote neutral posts, as indicated by the slight upward trend of the linear estimator for the promoted activity (m = 1 × 10^−4^), which is greater than the estimator of the companies’ activity (m = 2 × 10^−5^). Finally, the emotions of anger, disgust, sadness, and surprise were detected with significantly lower values; the corresponding figures are included in Figure A1 of Appendix B.

### 3.4. Emotions Triggered in Users by Pharmaceutical Companies’ Posts in the COVID-19 Channel (RQ3)

Table 8 presents the emotional profiles of the public’s reactions. When users actively responded to the pharmaceutical companies with a quote or a reply, they preferred negatively charged content compared to the pharmaceutical companies’ posts, since all emotions of a negative nature are enhanced. The emotion of fear is increased in the public’s response, both in quotes (0.1463) and in replies (0.1830), compared to the values of the companies’ activity in the COVID-19 channel (0.0778 in Table 7). The average levels of anger, disgust, and sadness are noticeably increased compared to the corresponding average values in pharmaceutical companies’ posts but are still restricted to low values (i.e., <0.07). Similarly, the levels of surprise are elevated from 0.0658 in pharmaceutical companies’ posts to 0.2619 and 0.2603 in users’ quotes and replies while, on the other hand, the values of joy are almost halved, from 0.4473 in the COVID-19 channel to 0.2830 and 0.1779 in the public’s response.

On a second level of analysis, we focused on the public’s response to other users’ quotes and replies to highlight the interactions amongst the recipients of the pharmaceutical companies’ posts. Through users’ retweet behavior, we highlighted the endorsement or rejection of other users’ opinions by computing the weighted average values of emotions based on the public’s retweets and comparing them to the corresponding values of the posts that triggered this reaction. Table 8 indicates that users clearly favored quotes and replies that conveyed joy or were neutrally charged and refrained from sharing negatively charged posts. The levels of joy were increased by 8.12% in the case of quotes and by 11.00% in replies, and neutral posts were enhanced by 61.29% and 29.05%, respectively. The values of anger, disgust, and sadness were decreased, although their low values can be misleading in terms of extracting credible conclusions. In the case of surprise, users triggered a change of −30.30% in quotes and −20.69% in replies. The emotion of fear was decreased in retweet activity in terms of both quotes and replies by 16.66% and 9.07%, respectively. 

The results indicate a clear enhancement of negatively charged emotions (i.e., anger, disgust, fear, sadness) and a significant drop in the emotion of joy, even in the flow of posts promoted by user retweets. Neutrally charged posts are at the same levels as in pharmaceutical companies’ activity and the emotion of surprise is elevated, indicating the public’s confusion caused by the information flow in the COVID-19 communication channel.

### 3.5. Users’ Response to Posts with Different Emotional Charges in the COVID-19 Channel (RQ4)

The analysis of emotional classes of pharmaceutical companies’ original posts detected emotions of fear, joy, and surprise or neutrality, as shown in Table 9. However, the analysis identified an extremely low presence of anger, disgust, and sadness (i.e., 14, 0, and 13 posts in each class, respectively); thus, these emotions were not considered for further analysis.

The overall reaction for the emotions of fear, joy, and surprise, presented in Table 9, was found to be significantly higher than the corresponding average values of the public’s general response to pharmaceutical companies’ posts in the COVID-19 channel (Table 6), signifying the public’s preference for promoting emotionally charged posts. Another key finding is the increased circulation of the fear class, as it exhibits the highest indicators in almost all categories. The average rate of likes for posts including fear is 127.48, a value noticeably higher than the corresponding values of other classes, indicating the public’s tendency to endorse messages that convey fear, while the average retweet rate is 29.61, a finding that supports our observation about the increasing trend in the emotion of fear, due to the enhanced promotion of posts in this class via user retweets, as noticed in Figure 5b. Posts in the surprise class exhibit the second highest indicator values, with a clear distance from the fear class, and posts indicative of joy have the third highest total activity and the highest quote rate. The class of neutral posts exhibits the lowest indicator values, demonstrating, thus, the public’s indifference towards posts devoid of emotional charge.

The results presented in Table 10 indicate a noticeable inclination of the public to maintain high levels of the dominant emotion in each class (i.e., the grouping emotion of each class, as indicated by shaded values in Table 10). As denoted by the downwards arrows, the average values of each class’s dominant emotion are either at the same levels or marginally decreased (i.e., <10^−2^) in the classes of joy, surprise, and neutral charge, while, in the case of fear, the average value of user promotion is significantly lower, indicating, thus, the public’s tendency to circulate posts with a lower charge of fear than the average. Furthermore, the emotions of fear and surprise are increased in almost every class, except for the cases when they are the dominant emotion.

## 4. Discussion

A key finding of our analysis is the dominance of the emotion of joy in both channels throughout the period of 31 months under study, followed by the presence of a neutral emotional charge in the posts, as highlighted by the results of RQ1. When the response of the public is examined (RQ2), joy was the emotion with the highest circulation based on the retweet rate of the public. The negative emotion of fear was in the third place, with the second highest retweet rate, followed by surprise, while the emotions of anger, disgust, and sadness had low values that cannot support the extraction of credible conclusions. 

The analysis of the public’s response in terms of like, reply, quote, and retweet rates in the COVID-19 communication channel revealed an increased level of activity by users and, consequently, their expected interest in the information disseminated by scientific beacons in times of pandemic. Similar findings are reported in [43], where users exhibited distinctively higher rates of likes and retweets towards German experts compared to authoritative users. Activity in the COVID-19 communication channel, combined with the confirmed high retweet rates that boosted the circulation of posts indicative of fear, joy, and surprise, as revealed by the findings of RQ4, is a clear signal for the inclination of the public towards posts with an intense emotional charge and can serve as a guideline for the development of efficient information diffusion strategies in the future.

Posts made by the pharmaceutical companies in the COVID-19 channel resulted in tweets with an elevated negative emotional charge, as concluded by the results of RQ3. Users produced quotes and replies with joy as the predominant emotion but with a starker presence of fear and sadness compared to the emotional profile of the pharmaceutical companies’ posts. Although the promotion of user-to-user quotes and replies moderated negative emotions and enhanced the emotion of joy, the overall average emotional charge was different than the profile of the channel and was negatively inclined.

The emotional profile of the COVID-19 communication channel is dominated by joy, followed by a neutral charge and a striking, yet not dominant, presence of fear and surprise. This profile is in accordance with other studies that examine the public’s attitude towards COVID-19 and, more specifically, vaccine-related topics on social networks. In [20], the authors conclude that positive sentiments dominate, followed by neutral content and a much smaller portion of negative sentiment, while in [49], the results highlight an emerging trend of negative sentiment in successive lockdowns in India. In [25,26,27,31], the findings indicate that the public most frequently exhibits positive emotions (e.g., trust) towards vaccines, followed by a mixture of negative emotions (e.g., fear, sadness, anger), and in [21], the authors detect marginal fluctuations between positive and negative posts in between vaccination calls. 

The prevalence of joy is in agreement with the findings of other studies that harvest data from Twitter or other micro-blogging platforms [22]. Studies that focus on narrow periods in the beginning of the pandemic [23,50] as well as works covering wider periods [24,51,52,53,54] affirm the dominance of joy despite some early outbursts of negatively charged posts. The prevalence of joy is also supported by studies investigating the acceptance of the newly developed vaccines that utilized data from adjacent or even overlapping communication channels [55,56].

The results of this work indicate a stark presence of fear in the public’s response in the COVID-19 communication channel, with an upward trend. Recent studies support that negative sentiments are dominant on social media in the COVID-19 context [48,57] and detect fear and even panic as the prevailing emotion [28,58,59]. Although our findings do not provide grounds for similar conclusions, the steadily increasing presence of fear was detected in users’ responses. These deviations may be attributed to the different datasets that are processed in the analyses.

## 5. Conclusions

Our analysis of the COVID-19 and non-COVID-19 communication channels revealed the predominance of joy in posts made by pharmaceutical companies in both channels, followed by tweets with a neutral charge, while the emotions of surprise and fear were present with lower values. The public responded more intensely to posts in the COVID-19 channel, which is evidence of the public’s interest in the information disseminated by credible sources, but the emotional profile of their responses was negatively inclined. The level of joy was decreased compared to the emotional profile of the companies’ activity, although it was still the predominant emotion, while fear and sadness were enhanced. 

The analysis of the diffusion indicators (i.e., rates of likes, quotes, replies, and retweets) highlighted the public’s tendency to more actively promote posts in the COVID-19 channel. Tweets conveying an intense emotional charge exhibited high rates of circulation, with joy and fear sharing the two highest rates. Users’ quotes and replies to pharmaceutical companies’ posts differed from the average emotional profile of the COVID-19 channel and indicated an enhancement of fear and a restraint in the levels of joy.

Our findings highlight the contribution of emotional charges to posts disseminated by credible sources in times of peril and their catalytic role in the posts’ endorsement and diffusion by the public. These results provide grounds for the improvement of communication strategies by actors in future crises through the deployment of emotion-aware messages. 

### Limitations and Future Directions

One of the main limitations of this study is the emotion analysis scheme we used, which was limited to the six basic emotions included in the Ekman classification, where four of the emotions are negatively charged (i.e., anger, disgust, fear, sadness), while alternative approaches propose broader schemes of eight [58] or sixteen emotions [60]. An enriched set of emotions would have fueled a more fine-grained analysis of online activity. Another limitation stems from the selection of a single emotion analysis technique, coupling our results with the performance of the specific algorithmic approach. An ensemble of state-of-the-art algorithmic tools with aggregation rules would have strengthened the robustness of the presented methodology and minimized the risk of misclassifying a tweet due to the poor performance of a single algorithm. Similarly, the distinction between the COVID-19 and non-COVID-19 communication channels was based on the annotation of the Twitter platform; the utilization of topic detection techniques could have provided a different and, possibly, more descriptive partition. Our approach focused on the emotional investigation of interactions between the public and pharmaceutical companies. A more analytic approach could have revealed special features of the exchanged messages and patterns in the tweet–retweet–reply discourse. Finally, this study exploited posts related to COVID-19 only written in English, while a multilingual approach would have revealed different findings in diverse cultural settings.

## Figures and Tables

**Figure 1 behavsci-14-00128-f001:**
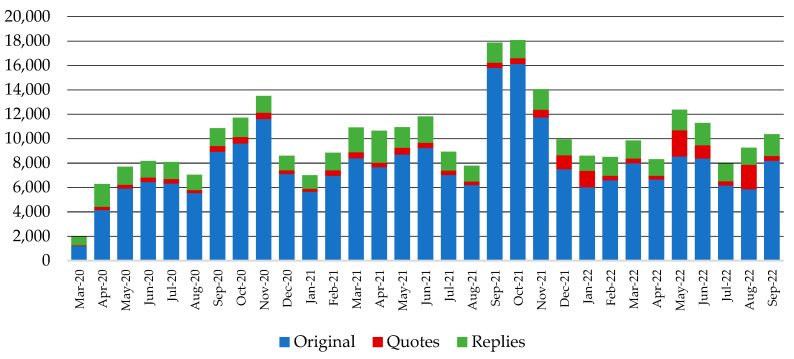
Pharmaceutical companies’ activity in the non-COVID-19 communication channel for the period between March 2020 and September 2022.

**Figure 2 behavsci-14-00128-f002:**
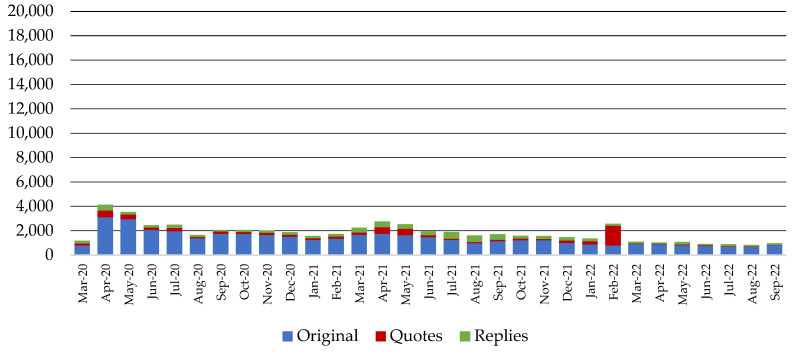
Pharmaceutical companies’ activity in the COVID-19 communication channel for the period between March 2020 and September 2022.

**Figure 3 behavsci-14-00128-f003:**
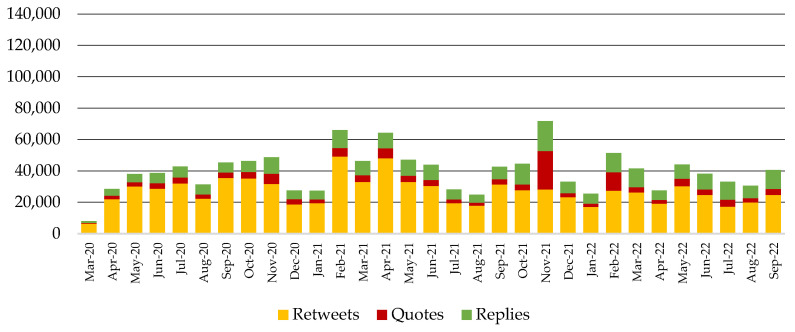
Users’ activity in the non-COVID-19 communication channel triggered by companies’ original posts, quotes, and replies for the period between March 2020 and September 2022.

**Figure 4 behavsci-14-00128-f004:**
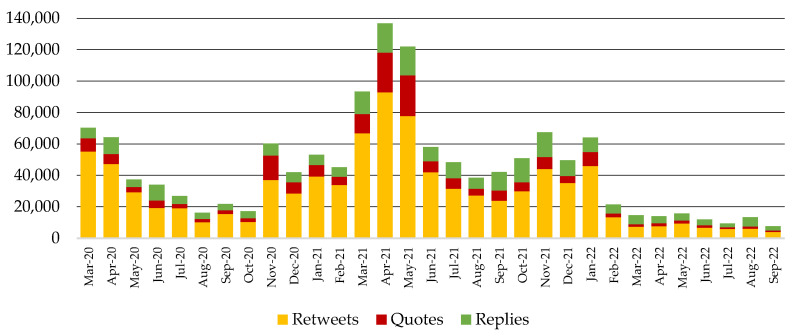
Users’ activity in the COVID-19 communication channel triggered by companies’ original posts, quotes, and replies for the period between March 2020 and September 2022.

**Figure 5 behavsci-14-00128-f005:**
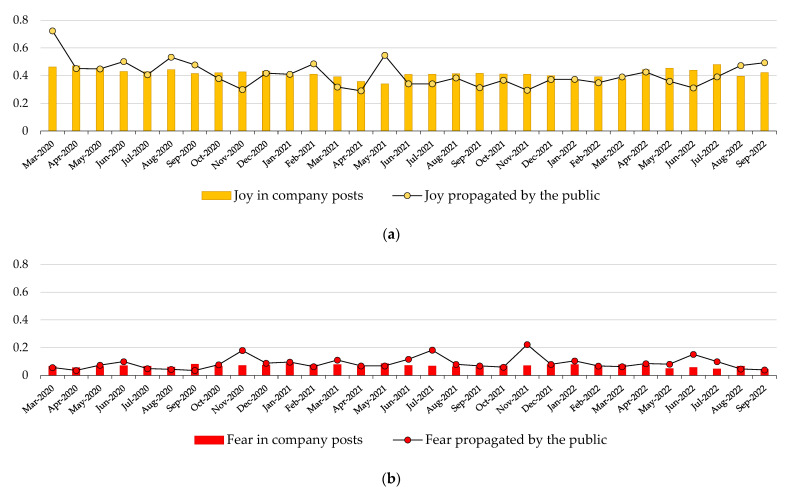
Average emotions in pharmaceutical companies’ posts per month and posts retweeted by the public throughout the period under study, with a focus on (**a**) joy, (**b**) fear, and (**c**) neutrally charged posts.

**Table 1 behavsci-14-00128-t001:** Dataset of pharmaceutical companies’ activity in the non-COVID-19 and the COVID-19 communication channels.

Pharmaceutical Companies’ Activity	Non-COVID-19		COVID-19	
	Total	English	Total	English
Original tweets	241,971	156,052	41,886	27,723
Retweets	40,132	24,806	8413	6054
Replies	47,286	29,364	7789	5925
Quotes	17,827	12,477	7048	5073
Total interactions	347,216	222,699	65,136	44,775

**Table 2 behavsci-14-00128-t002:** The public’s response to pharmaceutical companies’ posts (i.e., original tweets, replies, and quotes) in the non-COVID-19 and COVID-19 communication channels.

Public’s Response	Non-COVID-19	COVID-19
Retweets	828,552	921,121
Replies	262,838	250,578
Quotes	135,993	195,357
Total interactions	1,227,383	1,367,056

**Table 3 behavsci-14-00128-t003:** Retrieved quotes and replies made in English by the public to pharmaceutical companies’ English-language posts (i.e., original tweets, replies, and quotes) in the COVID-19 communication channel.

Retrieved Public’s Response (In English)	COVID-19
Replies	195,597
Quotes	60,650
Total	256,247

**Table 4 behavsci-14-00128-t004:** Emotional profiles of the pharmaceutical companies’ English posts (i.e., original posts, quotes, and replies) in the non-COVID-19 and the COVID-19 channel.

Channel	Number of Posts	Anger	Disgust	Fear	Joy	Sadness	Surprise	Neutral
non-COVID-19	197,904	0.0072	0.0023	0.0642	0.4583	0.0102	0.0949	0.3629
COVID-19	38,721	0.0081	0.0046	0.0778	0.4053	0.0148	0.0694	0.4199

**Table 5 behavsci-14-00128-t005:** Response ratio of the public’s actions (i.e., retweets, replies, quotes) to pharmaceutical companies’ posts (i.e., original posts, quotes, replies).

Channel	Pharmaceutical Companies’ Tweets	Public’s Response	Response Ratio
non-COVID-19	307,084	1,227,383	4.00
COVID-19	56,723	1,367,056	24.10

**Table 6 behavsci-14-00128-t006:** Users’ average likes, quotes, replies, and retweets per pharmaceutical companies’ post type in the non-COVID-19 and COVID-19 communication channels.

Channel	Type	Average Rates				
		Like	Quote	Reply	Retweet	Total
non-COVID-19	Original	18.37	0.53	0.96	3.20	23.06
	Quote	5.30	0.11	0.33	1.49	7.23
	Reply	3.38	0.10	0.53	0.60	4.61
COVID-19	Original	80.60	4.35	5.46	19.95	110.35
	Quote	25.48	0.65	1.15	6.11	33.39
	Reply	22.01	1.09	1.77	5.47	30.33

**Table 7 behavsci-14-00128-t007:** Emotional profiles of the non-COVID-19 and COVID-19 channels of pharmaceutical companies’ posts (repeated from Table 4 for comparison purposes) and their enhancement by the public through retweet activity (changes >10^−2^ are highlighted with indicative arrows).

Channel	Emotional Profile	Anger	Disgust	Fear	Joy	Sadness	Surprise	Neutral
non-COVID-19	Pharmaceutical companies	0.0072	0.0023	0.0642	0.4583	0.0102	0.0949	0.3629
	Users’ promotion	0.0068	0.0019	↑0.0788	↓0.4206	0.0105	0.0923	↑0.3890
COVID-19	Pharmaceutical companies	0.0081	0.0046	0.0778	0.4053	0.0148	0.0694	0.4199
	Users’ promotion	0.0044	0.0028	↑0.0927	0.4093	0.0109	↑0.0889	↓0.3910

**Table 8 behavsci-14-00128-t008:** The emotional profiles of users’ quotes and replies and changes due to users’ promotion through retweets (changes >10^−2^ are highlighted with indicative arrows).

Public’s Reaction	Anger	Disgust	Fear	Joy	Sadness	Surprise	Neutral
Quotes	0.0296	0.0115	0.1463	0.2830	0.0630	0.2619	0.2046
% change due to retweets	↓−47.29%	↓−41.97%	↓−16.66%	↑+08.12%	↓−40.94%	↓−30.30%	↑+61.29%
Retweeted quotes	0.0156	0.0067	0.1220	0.3060	0.0372	0.1826	0.3300
Replies	0.0307	0.0135	0.1830	0.1779	0.0695	0.2603	0.2651
% change due to retweets	↓−21.54%	↓−16.97%	↓−09.07%	↑+11.00%	↓−24.77%	↓−20.69%	↑+29.05%
Retweeted replies	0.0241	0.0112	0.1664	0.1975	0.0523	0.2064	0.3421

**Table 9 behavsci-14-00128-t009:** Number of posts and average rates of likes, quotes, replies, retweets, and all actions by users on posts in the detected classes (classes with statistical uncertainty are not included). Highest values in each reaction category are highlighted with bold.

Post Class	Number of Posts	Average Rates				
		Like	Quote	Reply	Retweet	Total
Fear	1307	**127.48**	4.30	**8.05**	**29.61**	**169.44**
Joy	14,224	82.63	**5.27**	5.70	20.89	114.49
Surprise	1177	102.01	3.98	6.07	22.40	134.45
Neutral	22,027	77.84	4.06	5.66	18.45	106.01

**Table 10 behavsci-14-00128-t010:** Emotional profiles of pharmaceutical companies’ posts per class and users’ promotion through retweet activity (changes >10^−2^ are highlighted with indicative arrows).

Class of Posts	Source	Number of Posts	Average Circulated Emotion						
			Anger	Disgust	Fear	Joy	Sadness	Surprise	Neutral
Fear	Pharmaceutical companies	1307	0.0126	0.0076	0.6577	0.0691	0.0108	0.0575	0.1848
	Users’ retweets	38,695	0.0129	0.0089	↓0.5723	0.0781	0.0120	↑0.0925	↑0.2233
Joy	Pharmaceutical companies	14,224	0.0008	0.0001	0.0108	0.7478	0.0018	0.0272	0.2115
	Users’ retweets	297,141	0.0016	0.0004	↑0.0209	0.7534	0.0042	↑0.0445	↓0.1750
Surprise	Pharmaceutical companies	1177	0.0138	0.0143	0.1277	0.1361	0.0331	0.6387	0.0363
	Users’ retweets	26,362	0.0151	0.0158	↑0.1438	↑0.1465	0.0343	↓0.6123	0.0321
Neutral	Pharmaceutical companies	22,027	0.0118	0.0067	0.0847	0.2189	0.0222	0.0673	0.5885
	Users’ retweets	406,320	0.0049	0.0032	↑0.0967	↓0.2066	0.0139	↑0.0871	0.5877

## Data Availability

All data and analysis methodologies are contained in the manuscript. Any additional data requests can be addressed to the corresponding authors.

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
