# Peer review of "Analysis of Pharmaceutical Companies’ Social Media Activity during the COVID-19 Pandemic and Its Impact on the Public"

_behavsci, 2024, doi:10.3390/bs14020128_

Round 1
Reviewer 1 Report
Comments and Suggestions for Authors
The study aims to compare the companies' communication in the COVID-19 and non-COVID-19 channels, analyse the emotional profile of their posts and examine the public's response. Additionally, the research seeks to understand the public's response to the companies' posts and to shed light on communication strategies for future crises. It addresses a specific gap in the field by examining the communication strategies of pharmaceutical companies during a global health crisis and their impact on public sentiment.
The study delves into the public's preference for posts with different emotional profiles, the evolution of emotions in the COVID-19 channel, and the impact of emotional charge on the endorsement and diffusion of posts. Additionally, the research provides insights into the most frequently used hashtags in posts by pharmaceutical companies and promoted posts by users, offering a detailed understanding of the topics and themes being discussed on Twitter during the pandemic. Overall, the article contributes valuable insights into the use of social media for public health research, sentiment analysis, and the spread of information during a global health crisis, which is a significant addition to the subject area compared with other published material.
The authors should consider providing more details about the process of dataset compilation, particularly regarding the criteria used for the detection and validation of appropriate Twitter accounts of pharmaceutical companies. This would enhance the transparency and reproducibility of the study. Additionally, the authors could consider incorporating formal criteria for the detection and validation of appropriate accounts, which would further strengthen the credibility of the retrieved data. Furthermore, the authors may want to consider including additional controls for the sentiment and emotion analysis, such as comparing the results with those obtained from other sentiment analysis techniques or emotion analysis algorithms to ensure the robustness of their findings.
The conclusions drawn in the investigation are consistent with the evidence and arguments presented. Furthermore, they are supported by the evidence presented throughout the investigation. However, it would be good if a paragraph indicating the limitations of the study were included. Only in the "Results" section is there mention of a limitation of the study, but this should be highlighted in the conclusions as well.
The references provided are appropriate and relevant to the discussion. They support the analysis and findings presented in the research, including the compilation of the dataset, the analysis of public response, and the tools and techniques used for the study. These references contribute to the credibility and robustness of the research by providing context and supporting evidence for the methodologies and conclusions.
Reviewer 2 Report
Comments and Suggestions for Authors
This research is very interesting and presents very relevant results. Overall, the article is well written, but could be improved for publication. I therefore consider that
1) the theoretical framework could be enriched by exploring the importance of social media at the time of the pandemic as a means of communication/information with the public;
2) the research design and methodological options should be better justified: why did they choose Twitter/X and not another social network; what is the difference between Covid-19 and non-Covid-19 communication channels? what is the model that uses a recurrent neural network (RNN)?
3) we are presented with research questions, but not clear objectives. It would be pertinent to clarify what the objectives are, so that the discussion of the results and conclusions can then systematise how the objectives were or were not achieved.
4) How did you process the data statistically? What software did you use?
Finally, the interpretation of the data presented in the tables and graphs could be made clearer and more concise.
Comments on the Quality of English Language
A brief revision of the text is needed to eliminate some translation problems.
